# Dual-Polarization Radar-Based Quantitative Precipitation Estimation of Mountain Terrain Using Multi-Disdrometer Data

**Cheol-Hwan You [1], Sung-Ho Suh [2], Woonseon Jung [3], Hyeon-Joon Kim [4,*] and Dong-In Lee [1]**

1   Atmospheric Environmental Research Institute (AERI), Pukyong National University (PKNU), 45, Yongso-ro, Nam-gu, Busan 48513, Korea; youch@pknu.ac.kr (C.-H.Y.); leedi@pknu.ac.kr (D.-I.L.)
2   Flight Safety Technology Division, NARO Space Center, Korea Aerospace Research Institute (KARI), 508, Haban-ro, Bongrae-myeon, Goheung-gun 59571, Korea; suhsh@kari.re.kr
3   Convergence Meteorological Research Department, National Institute of Meteorological Sciences (NIMS), Jeju 63568, Korea; wsjung01@korea.kr
4   Department of Civil and Environmental Engineering, College of Engineering, Chung-Ang University, 84, Heukseok-ro, Dongjak-gu, Seoul 06974, Korea
*   Correspondence: hjkim22@cau.ac.kr; Tel.: +82-010-6566-5579

**Abstract:** The precipitation systems that pass over mountains develop rapidly due to the forcible ascent caused by the topography, and spatial rainfall distribution differences occur due to the local development of the system because of the topography. In order to reduce the damage caused by orographic rainfall, it is essential to provide rainfall field data with high spatial rainfall accuracy. In this study, the rainfall estimation relationship was calculated using drop size distribution data obtained from 10 Parsivel disdrometers that were installed along the long axis of Mt. Halla (oriented west–east; height: 1950 m; width: 78 km; length: 35 km) on Jeju Island, South Korea. An ensemble rainfall estimation relationship was obtained using the HSA (harmony search algorithm). Through the linear combination of the rainfall estimation relationships determined by the HSA, the weight values of each relationship for each rainfall intensity were optimized. The relationships considering $K_{DP}$, such as $R(K_{DP})$ and $R(Z_{DR}, K_{DP})$, had higher weight values at rain rates that were more than 10 mm h$^{-1}$. Otherwise, the $R(Z_H)$ and $R(Z_H, Z_{DR})$ weights, not considering $K_{DP}$, were predominant at rain rates weaker than 5 mm h$^{-1}$. The ensemble rainfall estimation method was more accurate than the rainfall that was estimated through an independent relationship. To generate the rain field that reflected the differences in the rainfall distribution according to terrain altitude and location, the spatial correction value was calculated by comparing the rainfall obtained from the dual-polarization radar and AWS observations. The distribution of Mt. Halla's rainfall correction values showed a sharp difference according to the changes in the topographical elevation. As a result, it was possible to calculate the optimal rain field for the orographic rainfall through the ensemble of rainfall relationships and the spatial rainfall correction process. Using the proposed methodology, it is possible to create a rain field that reflects the regional developmental characteristics of precipitation.

**Keywords:** quantitative precipitation estimation; dual-polarization radar; disdrometer; orographic precipitation; spatial bias correction; heuristic optimization algorithm

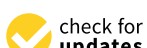



## 1. Introduction

High-impact weather, such as localized flash floods in mountainous terrain, can cause damage to a small area within short amounts of time, ranging from several minutes to tens of minutes. Rainfall information that is based on ground AWS (automatic weather system) observations have limitations in providing accurate rainfall information due to the gaps in ground observation sites and the time resolution of the observation data. To solve the limitations of obtaining extreme weather information based on in situ observations, it is necessary to observe a wider area of precipitation simultaneously. To calculate rainfall distribution with a high temporal resolution of a few minutes over wider areas of 200 km

or more, remote sensing data, such as weather radar and satellite data, are used for quantitative prediction estimation (QPE).

Many technological and academic advances have been made, such as advancements in observation equipment, improvements in relational expressions based on long-term observation data, and rainfall optimization according to locational characteristics [1–4]. However, there are still uncertainties due to the conversion of the rainfall intensity determined from reflectivity and beam blocking due to the fact of terrain and buildings [5]. In addition, due to the differences in the spatio-temporal resolution between the rainfall in the upper layer observed through the radar and the rainfall obtained through in situ surface observation data, the rainfall data based on radar observations contain errors in terms of the spatio-temporal resolution [6,7].

To reduce the uncertainty of rainfall estimation and to improve the accuracy of rainfall estimations based on dual-polarization radar observation data, research on how to improve accuracy through a combination of dual-polarization parameter characteristics [8–12], how to improve accuracy by considering beam blocking [13–15], and how to correct deviations based on the distance and altitude of radar beams [16,17] has been conducted. Furthermore, rainfall information that is suitable for determining rainfall characteristics and geographic conditions is able to be calculated using the results from previous research.

However, the methods mentioned earlier are often adjusted to the climatic precipitation characteristics or regional and seasonal characteristics based on long-term observational data and have limitations in that they cannot consider the complexity of precipitation. As a result, differences in the precipitation development characteristics may appear for each elevation, region, and precipitation cell within the precipitation system, and differences in the rainfall estimation accuracy for small spatial scales may occur. Therefore, to generate representative precipitation data, (1) an approach using an estimated relational ensemble [11,18,19] and (2) the correction of spatial rainfall distribution errors [20] were conducted.

Li et al. [18] obtained relationships through combining dual-polarization parameters (i.e., $Z_H$, $Z_{DR}$, and $K_{DP}$) and a probabilistic approach with a Gaussian mixture model for several estimation expressions. They proposed that the GMRE (Gaussian mixture rainfall-rate estimator) is not restricted by the radar frequency range and is one of the solutions to improving rainfall accuracy with rainfall types and regional rainfall characteristics. The GMRE method has the advantage of reflecting the climatic characteristics of precipitation as a statistical estimation method based on long-term observation data. However, it is limited in that sufficient observation data must be secured.

You et al. [11] retrieved dual-polarization parameters (i.e., $Z_H$, $Z_{DR}$, $K_{DP}$, and $A_H$) using surface disdrometer (Parsivel and POSS) observation data collected over a long period of time in a coastal area, and the rainfall estimation relations were generated through a combination of dual-polarization parameters. The accuracy of each rainfall estimation relationship was examined using ground rain gauge observation data. In addition, the rainfall estimation relationship was created through a linear combination of the generated rainfall estimation relationships. The weight of the linear combination was calculated through the error value (RMSE). It was found that by using a linear combination of rainfall estimation relationships, the ensemble method had higher accuracy than the independent rainfall estimation relationship results.

Kang et al. [20] performed spatial correction using the deviation value of rainfall collected through radar-based rainfall and ground observations and proposed a probabilistic solution to the uncertainty of radar rainfall through spatial correction. By calculating the error for each site of radar rainfall and ground rainfall and modeling the correlation of the rainfall error between the ground observation sites, a perturbation field reflecting the spatial error of the two-dimensional rainfall was created and applied to the radar rainfall fields. The spatial uncertainty of the radar rainfall was presented, and the spatial accuracy of the rainfall was improved by applying the observation error of each observation site to the radar rainfall field as a correction value.

Using the methods described above improved the rainfall estimation accuracy. However, since orographic precipitation systems over mountainous terrain have different microphysical characteristics, the rainfall accuracy is dependent on the altitude and location of the mountain.

QPE studies [21–23] in mountainous regions are continuously being conducted. Most of the QPE studies on orographic rainfall focus on correcting rainfall errors by correcting the dual-polarization parameters according to mountain altitude. The amount of QPE research considering regional microphysical characteristics is insufficient. Therefore, in this study, the rainfall estimation relationship was calculated using multi-disdrometer observation data collected from 10 sites in a mountainous region. The optimal estimation relationship was obtained using an ensemble approach through a linear combination of the relationships with the HSA (harmony search algorithm). The optimal rainfall fields of the mountainous area were created through the spatial correction of the rainfall fields obtained via the rainfall estimation relationships for each site. The contents of this paper are organized as follows: In Section 2, the data and methods used for our analysis are presented. In Section 3, we present the results of the ensemble of rainfall relations and the spatial correction for the orographic rainfall. In Section 4, the summary and conclusions of this study are presented.

## 2. Observational Data and Methodology

### 2.1. Intensive Field Observation Campaign

2.1.1. Disdrometer Observation Network on Jeju Island

To calculate the rainfall estimation relationship to generate the rainfall fields to determine the orographic rainfall, 10 Parsivel disdrometer instruments were installed along the long axis of Mt. Halla (Figure 1) during the rainy season every summer from 2012 to 2014 (Table 1). To consider the differences in raindrop development according to the altitude differences and the location of the mountainous terrain (windward/leeward), which are mainly influenced by the westerlies, the location of the Parsivel disdrometer observation site was set along the slopes of Mt. Halla and spanned from the southwest coastal area to the northeast coastal region of Jeju Island (Table 2). The maximum range of the dual-polarization radar observation was 240 km, and the data, obtained every 5 min operated by KMA (Korea Meteorological Administration) on the west (GSN) and east (SSP) coasts of Jeju Island, were used to apply the rainfall estimation relationship that was calculated using Parsivel disdrometer data and to generate two-dimensional rainfall fields. The spatial and the azimuthal resolutions of the dual-polarization radars were 250 m and 1.0°, respectively. The volume scans included 9 elevation angles (i.e., 0.2°, 0.5°, 1.0°, 1.6°, 2.4°, 3.5°, 5.2°, 7.6°, and 15.0°). Spatial correction and verification of the two-dimensional rainfall fields were performed using rainfall amount information obtained through AWS observations operated by KMA.

Information, such as rainfall intensity and the reflectivity of a set temporal resolution, can be calculated using the information on the diameter and fall speed of precipitation particles obtained during the Parsivel disdrometer observations. In this study, observation data set with a 1 min time resolution was used to consider the changes in precipitation characteristics over time. Particle diameters, ranging from 0.2 to 25 mm, and fall velocities, ranging from 0.2 to 20 m s$^{-1}$ (Table 3), can be observed by the Parsivel disdrometer [24]. There were 32 observation channels for both the particle diameter and fall speed; therefore, 1024 channels (32 × 32) were observed to determine the number of particles during the set time resolution, and the concentration value for each channel was calculated by considering the number of particles observed for each channel and the observation area and temporal resolution of the laser beam (Table 3). Among the 32 channels for the diameter information and fall speed information, the first and second channels were not included in the observable range of the Parsivel disdrometer and were treated as noise.

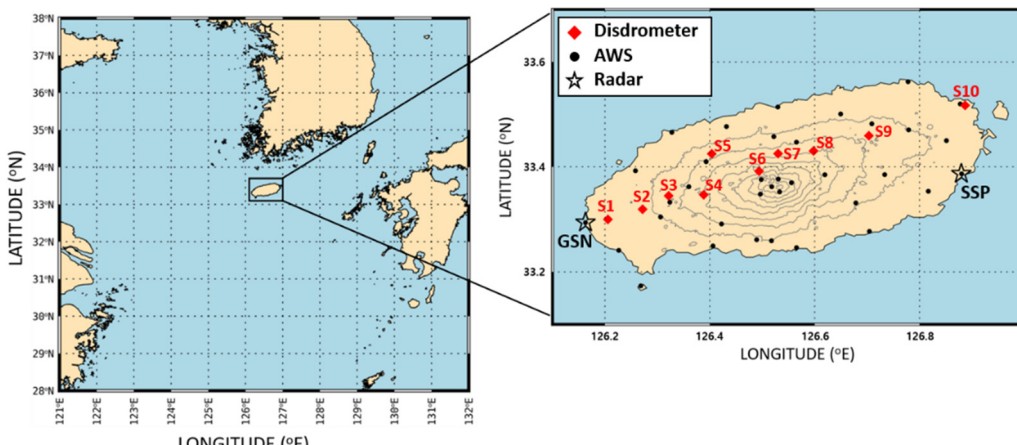

**Figure 1.** The location (red diamond) of the installed Parsivels. The two black open stars indicate the GSN (western) and SSP (eastern) radar sites. The black dots indicate the AWS sites installed by KMA. The thin gray contour lines show the topography (contour interval: 200 m).

**Table 1.** Parsivel- and rain gauge-based orographic rainfall observation periods during the summer rainfall season.

| Observation Year | Observation Period |
|------------------|--------------------|
| 2012 | 27 June–13 July |
| 2013 | 18 June–14 July |
| 2014 | 19 June–14 July |

**Table 2.** Location and altitude information from the Parsivel and rain gauge observation sites on Jeju Island.

| Site | Latitude (°, N) | Longitude (°, E) | Altitude (m) |
|------|-----------------|------------------|--------------|
| S1 | 33.3000 | 126.2056 | 37 |
| S2 | 33.1394 | 126.2717 | 140 |
| S3 | 33.3450 | 126.3214 | 324 |
| S4 | 33.3469 | 126.3883 | 551 |
| S5 | 33.4250 | 126.4036 | 330 |
| S6 | 33.3919 | 126.4939 | 975 |
| S7 | 33.4253 | 126.5303 | 571 |
| S8 | 33.4303 | 126.5978 | 590 |
| S9 | 33.4594 | 126.7033 | 332 |
| S10 | 33.5172 | 126.8869 | 57 |

**Table 3.** The specification information from the Parsivel disdrometer.

| Parsivel Disdrometer | | Technical Data |
|----------------------|--|----------------|
| Wavelength of the optical sensor | | 780 nm |
| Measuring area | | $180 \times 30$ mm (54 cm$^2$) |
| Measuring range | Particle size | 0.2–25 mm (32 channels) |
| | Particle velocity | 0.2–20 m s$^{-1}$ (32 channels) |
| Precipitation intensity | | 0.001–1200 mm h$^{-1}$ |
| Measurement interval | | 10 s to 60 min |
| Dimensions (H $\times$ W $\times$ D) | | $670 \times 600 \times 114$ mm |

2.1.2. Calculating the Number Concentration

Parsivel disdrometers provide information on the diameter and falling speed of the precipitation particles. Using disdrometer observation data, the concentration value for

each drop diameter was calculated for the set temporal resolution (in this study, it was set to 1 min). Before calculating the concentration, the following process was carried out for QC (quality control) purposes [25,26] on nonweather observation data such as fallen leaves and insects: (i) the first and second diameter channels with low signal intensities were excluded [24,27,28]; (ii) data for the times when the rain rate value calculated from the Parsivel observation data were less than 0.1 mm h$^{-1}$ were excluded; (iii) data with a diameter of 8 mm or more, which are difficult to judge as liquid rainfall particles, were excluded; (iv) using the relationship (Equation (1)) [25,29] of the terminal velocity and the diameter of the raindrops obtained in the laboratory while also considering the environment in which raindrops are not affected by external factors, such as wind, when falling, only the values that were included in the effective fall velocity range for each raindrop diameter were used.

In this study, the constant in Equation (2) was set as 0.6, as suggested by Freidrich et al. [26].

$$V(D) = 9.65 - 10.3 \exp(-0.6D) \tag{1}$$

$$|V_{measured} - V_{Ideal}| < CV_{Ideal} \tag{2}$$

where D is the diameter of the raindrop, and $V_{measured}$ and $V_{ideal}$ are the measured fall velocity and terminal velocity for each diameter channel, respectively.

### 2.1.3. Retrieving the Dual-Polarization Parameters with T-Matrix

To calculate the rainfall estimation relationship by considering the dual-polarization parameters (i.e., $Z_H$, $Z_{DR}$, and $K_{DP}$), the dual-polarization parameters were retrieved using the concentration data at each site. It is possible to calculate dual-polarization parameters, such as $Z_H$, $Z_{DR}$, $K_{DP}$, and $\rho_{HV}$s, by applying the raindrop distribution data with a 1 min time resolution to the T-matrix scattering simulation derived by Mishchenko et al. [30].

This study calculated the reflectivity parameters using the T-matrix scattering simulation program written by Leinonen [31]. To estimate the dual-polarization parameters using the T-matrix scattering simulation, conditions, such as the shape, slope, and temperature, of the rainfall particles must be given. The raindrop oblateness relationship (Equations (3)–(5)) proposed by Thurai et al. [32] was used.

$$\frac{b}{a} = 1.0 \text{ for } D_{eq} < 0.7 \text{ mm} \tag{3}$$

$$\frac{b}{a} = 1.173 - 0.5165D_{eq} + 0.4698D_{eq}^2 - 0.1317D_{eq}^3 - 8.5 \times 10^{-3}D_{eq}^4 \text{ for } 0.7 \leq D_{eq} \leq 1.5 \text{ mm} \tag{4}$$

$$\frac{b}{a} = 1.065 - 6.25 \times 10^{-2}D_{eq} - 3.99 \times 10^{-3}D_{eq}^2 + 7.66 \times 10^{-4}D_{eq}^3 - 4.095 \times 10^{-5}D_{eq}^4 \text{ for } 1.5 \text{ mm} < D_{eq} \tag{5}$$

In the relational equation, a and b are the lengths of the horizontal and vertical axes of the raindrops, and $D_{eq}$ is the diameter of the raindrops. The set temperature variable was assumed to be 20 °C, and the result was calculated. It was applied by taking a Gaussian distribution with an average canting angle of 0° for the raindrops and a canting angle width of 20°. In order to apply the rainfall estimation relationship to the S-band dual-polarization radar operated by the KMA in Jeju, the frequency information of the S-band was used as an input variable for the scattering simulation.

### 2.2. Operational Data

#### 2.2.1. Dual-Polarization Radar and HSR

Weather radar is a remote sensing instrument that estimates the precipitation intensity by calculating the reduced intensity of electromagnetic waves through the transmission and reception of electromagnetic waves. Therefore, when the electromagnetic waves were blocked by the mountain terrain and buildings between a radar antenna and a target,

such as a precipitation cloud, the precipitation intensity decreased and caused the beam blocking phenomenon to occur. Jeju Island has a mountainous terrain and a high altitude of approximately 2 km ASL (above sea level), and it is a region where beam blacking by topography occurs frequently. Using the elevation angle data that allow an elevation that is higher than the topographical elevation to be observed, the beam blocking effect caused by the topography can be reduced. Precipitation particles may be affected by evaporation, breakup, shift effects, etc., resulting in a difference in the rainfall between the elevation of a radar observation area and the ground rain gauge observation site. Therefore, to minimize rainfall estimation errors according to the altitude, it is necessary to generate rainfall field data using the precipitation observation data of the lowest altitude that is closest to the ground altitude.

To minimize the beam blocking effect caused by Mt. Halla on Jeju Island, we applied the HSR (hybrid scan reflectivity) technique proposed by Lee et al. [33] and Lyu et al. [34]. For each observation azimuth angle of the radar, a scan elevation angle that was higher than the topographic elevation was obtained, and the reflectivity value for the calculated elevation angle was used to calculate the rainfall. To determine the rainfall fields for the orographic rainfall, the rain rate distribution data of the lowest altitude layer consisting of the cylindrical coordinate system (i.e., altitude angle, azimuth angle, and distance) acquired at each GSN and SSP site was converted into an orthogonal Cartesian coordinate system (x, y). In the overlapping observation area of the two radars, a value with a higher rain rate was used. Figure 2 shows the lowest elevation numbers for the GSN and SSP.

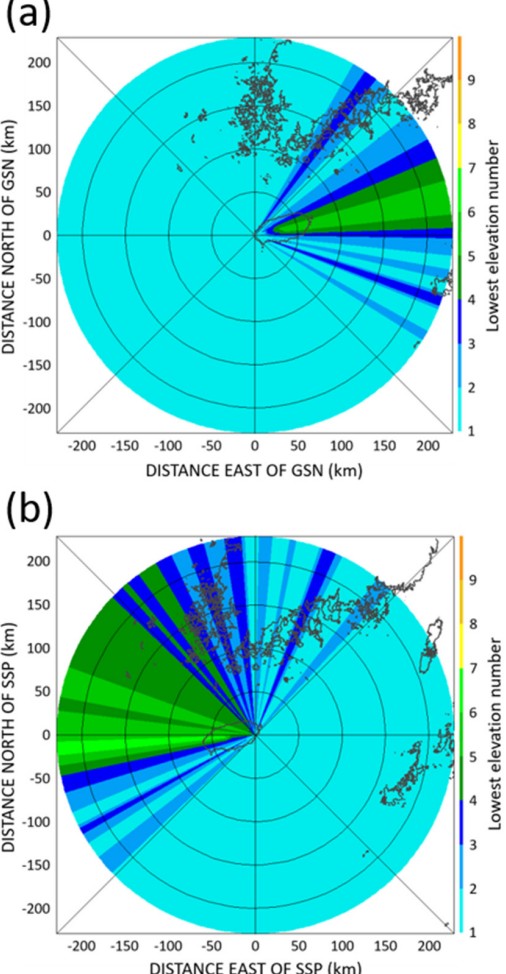

**Figure 2.** The lowest elevation angle with a beam elevation higher than the topographic elevation for each azimuth of (**a**) the GSN and (**b**) SSP S-band dual-polarization radars.

2.2.2. Ensemble of Rain Rate Relationships with HSA

The rainfall estimation relationship based on the ground disdrometer data has an error difference depending on the rainfall characteristics, and it is only possible to provide more accurate ground rainfall information when the rainfall estimation equation is generated through the optimal combination of dual-polarization parameters according to the microphysical characteristics of precipitation. Previously, many researchers have proposed the rainfall estimation relationship according to the geography and type of rainfall [35–37]. However, the rainfall estimation relationships suggested by previous researchers are limited in terms of their application in other regions and precipitation cases due to the differences in precipitation cases and observation instruments selected for calculating the relationships. In addition, unlike disdrometer observations in cities and flatlands, it is difficult to observe mountainous regions over a long period of time due to the geographical conditions. Therefore, in this study, to calculate the optimal rainfall estimation relationship for a precipitation system with various characteristics and that extends over mountainous terrain, the optimal rainfall estimation relationship was calculated through a linear combination of rainfall estimation relationships by considering several dual-polarization parameters. The optimization technique used to set the linear combination weight of the rainfall estimation relationships was the HSA (harmony search algorithm) [38–40].

The optimization technique used in the analysis is a valuable method when there is no correlation between the initial values used for optimization or when the solution of the equation shows a nonlinear pattern according to the changes in the parameter values. It is one of the metaheuristic optimization algorithms that is able to obtain the optimal solution through iterative calculations that mimic natural phenomena combinations without using an optimization method based on mathematical theory.

Linear weight values of randomly assigned rainfall estimation relationships are assigned, and the given weight value array is set as a number (in this study, 50 arrays were created). The RMSE (root mean square error) value for the ground AWS rainfall and the radar rainfall, to which several set weights were applied, were calculated. The RMSE (root mean square error) value of the ground AWS rainfall and the radar rainfall, to which several set weights were used, were calculated. The calculated RMSE value was the same as the previously set number of arrays. A new RMSE value was obtained by removing the row with the largest weight from the RMSE value array and by assigning a new random weight value. The optimal weight was calculated by repeating a random weight value generation and minimizing the RMSE. In this study, the number of repetitions was set to 20,000, and the optimum weight value was obtained by averaging the repetitive values of 20 sets.

2.2.3. Spatial Optimization of Rainfall Estimation

Since the rainfall field converted through the radar rainfall estimating equation is deterministic, to reduce the error caused by the spatial uncertainty, the error can be reduced through an ensemble that adds the uncertainty contained in the radar data [19]. The spatial ensemble of the radar rainfall field can be expressed by adding a spatial perturbation field to the radar rainfall field [40] (Equation (6)).

$$\Phi_{t,i} = R_t + \delta_{t,i} \tag{6}$$

where $\Phi_{t,i}$ denotes the probabilistic ensemble rainfall field; $R_t$ denotes the radar rainfall field for time $t$; $\delta_{t,i}$ represents the $i$th stochastic perturbation field based on the radar rainfall's temporal and spatial error for time $t$. The spatial characteristics of the radar rainfall can be considered by examining the ensemble results to which perturbation is applied. Yin et al. [41] mentioned that if the size of the spatial ensemble exceeds 10, the advantages of the ensemble are limited. The perturbation ($\delta_{t,i}$) representing the spatial error of the radar data is generated as the sum of the $Ly_{t,i}$ variables related to the correlation of the

ground observation site value at time $t$ to the mean error of the ground rainfall average observation error of the radar rainfall (Equation (7)).

$$\delta_{t,i} = \mu_{t,k} + Ly_{t,i} \tag{7}$$

where $\mu_{t,k}$ represents the average error from the rainfall start time to time $t$ at the observation site $k$ and is calculated by considering the difference between the ground observation rainfall value and the radar rainfall value (Equation (8)).

$$\mu_{t,k} = \frac{1}{\sum_{t=1}^{t} \omega_{t,x_k}} \omega_{t,x_k} \varepsilon_{t,x_k} \tag{8}$$

where $x_k$ represents the radar grid corresponding to the observation site $k$; $\varepsilon_{t,xk}$ and $\omega_{t,xk}$ are the observation error and the weight of the observation error at the observation sites $x_k$ and time $t$, respectively. In this study, the radar rainfall value was applied to the weight of the observation error [42]. $\varepsilon_{t,xk}$ can be expressed as Equation (9).

$$\varepsilon_t = 10 \log_{10}\left(\frac{G_t}{R_t}\right) \tag{9}$$

where $G_t$ and $R_t$ represent the ground rainfall values and radar rainfall values for the time $t$, and the unit of mm h$^{-1}$ was converted to dBR. To calculate $L$ in Equation (7), the covariance for each observation site should be calculated by considering the spatial variability of the radar rainfall (Equation (10)).

$$C = LL^T \tag{10}$$

where $C$ represents the covariance matrix between observation sites; and $L$ and $L^T$ represent the upper and lower triangular matrixes decomposed through the Cholesky algorithm ($L$ represents a lower triangular matrix). The covariance matrix ($C$) for each observation site can be calculated using Equation (11).

$$C_{kl} = \frac{1}{\sum_{t=1}^{Q} \omega_t, x_t} \sum_{t=1}^{Q} \omega_{t,x_k} \left(\varepsilon_{t,x_k} - \mu_k\right)\left(\varepsilon_{t,x_l} - \mu_l\right) \tag{11}$$

where $C_{kl}$ is the covariance value between the $k$ and $l$ observation sites; $Q$ is the time step. Assuming that the rainfall distribution has a time correlation [43], Equation (7) can be expressed as Equation (12).

$$\delta_{t,i} = \mu_{t,k} + v\delta'_{t,i} \tag{12}$$

Equation (12) is a relational expression that is converted by applying the AR(2) filtering model [44] to Equation (7) and is obtained through Equations (13)–(16).

$$\delta'_{t,i} = Ly_{t,i} - a_1\delta_{t-1,i} - a_2\delta_{t-2,i} \tag{13}$$

$$a_1 = r_1 \frac{r_2 - 1}{1 - r_1^2} \tag{14}$$

$$a_2 = \frac{r_1^2 - r_2}{1 - r_1^2} \tag{15}$$

$$v = \left[\frac{1 - a_2}{(1 - a_2)(1 - a_1 + a_2)(1 + a_1 + a_2)}\right]^{-0.5} \tag{16}$$

where $\alpha_1$ and $\alpha_2$ are the AR(2) parameters estimated by the Yule–Walker equations; $v$ is a scaling factor of AR(2) [42]. In Equation (13), $\delta'_{t,i}$ is a perturbation field with autocorrelation for two times $t - 2$ and $t - 1$.

## 3. Results

### 3.1. Rainfall Estimation Relationships with Dual-Polarization Radar Parameters

Using the raindrop size distribution data collected at the 10 Parsivel observation sites, the rainfall estimation relationship considering dual-polarization radar parameters (i.e., $Z_H$, $Z_{DR}$, and $K_{DP}$) was obtained through T-matrix scattering simulations. When calculating the dual-polarization parameters, the frequency information of the GSN (2825 MHz) and SSP (2755 MHz) dual-polarization radars was applied. Figure 3 is a scatterplot for the rain rate calculated using the rainfall estimation relations obtained through the concentration distribution data obtained at the S1 site and the rain rate calculated through the concentration data. The accuracy comparison variables of the rainfall estimation relationship were RMSE (root mean square error), NE (normalized error), and CORR (correlation coefficient).

$$\text{RMSE} = \left[ \frac{1}{N} \sum_{i=1}^{N} (R_{est} - R_{obs})^2 \right]^{1/2} \tag{17}$$

$$\text{NE} = \frac{\frac{1}{N} \sum_{i=1}^{N} |R_{est} - R_{obs}|}{\overline{R_{obs}}} \tag{18}$$

$$\text{CORR} = \frac{\sum_{i=1}^{N} (R_{est} - \overline{R_{est}}) (R_{obs} - \overline{R_{obs}})}{\left[ \sum_{i=1}^{N} (R_{est} - \overline{R_{est}})^2 \right]^{1/2} \left[ \sum_{i=1}^{N} (R_{obs} - \overline{R_{obs}})^2 \right]^{1/2}} \tag{19}$$

where $R_{est}$ and $R_{obs}$ are the estimated rain rate from the rainfall estimation relationship and the observed rain rate, respectively.

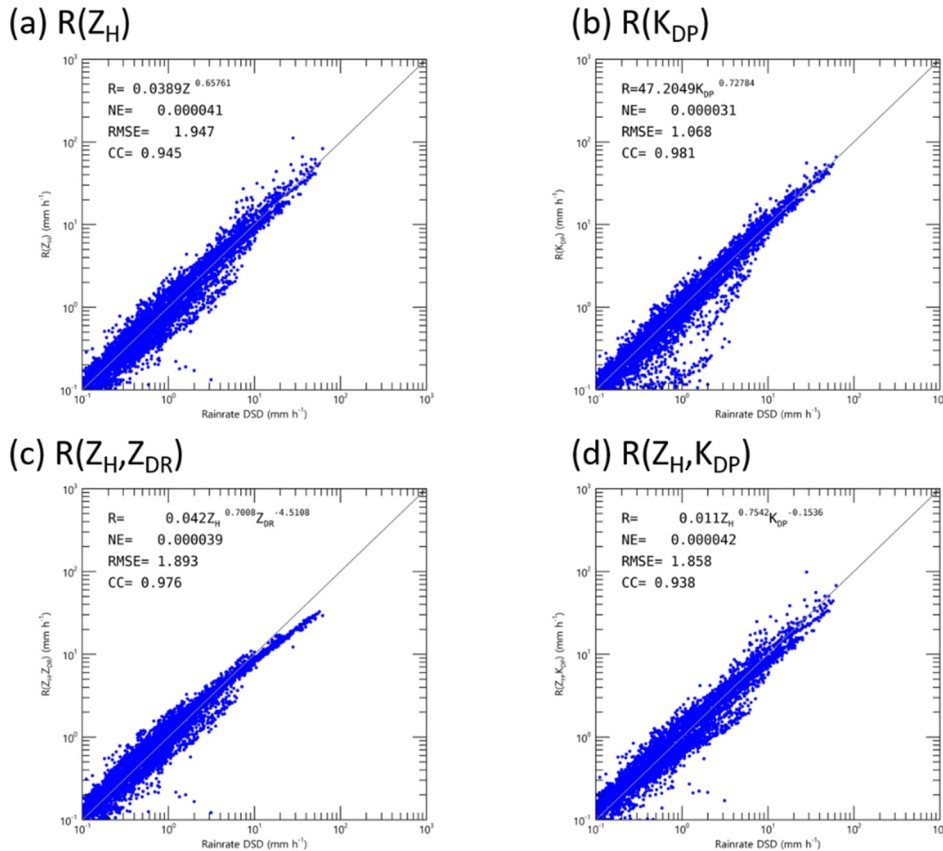

**Figure 3.** *Cont.*

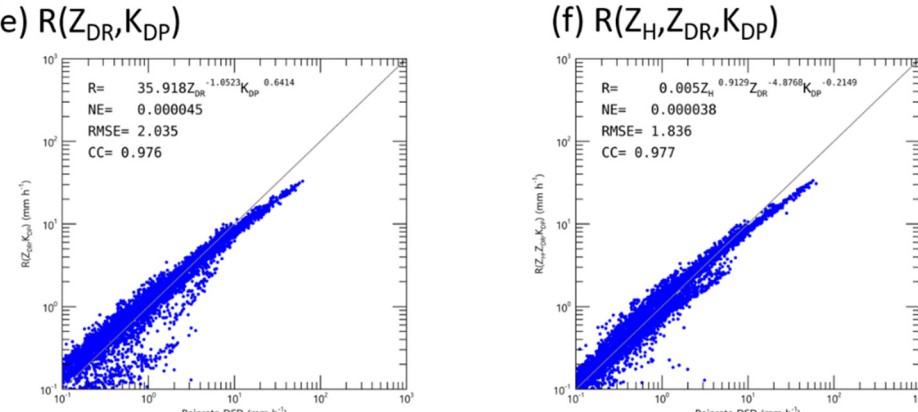

**Figure 3.** Scatterplot of the rain rates directly calculated from the DSDs versus those calculated from (**a**) $R(Z_H)$; (**b**) $R(K_{DP})$; (**c**) $R(Z_H, Z_{DR})$; (**d**) $R(Z_H, K_{DP})$: (**e**) $R(Z_{DR}, K_{DP})$; (**f**) $R(Z_H, Z_{DR},$ and $K_{DP})$ for GSN S-band radar simulated from Parsivel measurements at S1.

Tables A1 and A2 show the rainfall estimation relationships at each site. The RMSE (root mean square error) of the rainfall estimation relationship obtained at all sites, except S09, had values that were smaller than 3 mm h$^{-1}$ and CORR (correlation coefficient) values larger than 0.89. The RMSE and CORR of the rainfall estimation relationship at S09 that did not consider $Z_{DR}$ showed similar values to those of other sites. On the other hand, when $Z_{DR}$ was considered, the maximum RMSE was approximately 6.1 mm h$^{-1}$, and the minimum CORR value was 0.8. Rainfall fields were generated by applying the rainfall estimation relationships that were obtained via the in situ disdrometer observation data at each site to actual GSN and SSP dual-polarization radar data. To minimize the error with the ground rainfall value, the lowest altitude data that did not result in a beam block due to the influence of the terrain were used (Figure 2). To avoid considering non-meteorological echoes when generating rain rate field data, only grid data with $\rho_{HV}$ values of 0.85 or higher were used.

### 3.2. Optimization of Rainfall Estimate with the Ensemble Approach

Using the rainfall fields that were obtained, the optimal estimation relationship was calculated using a linear combination of rainfall estimation relationships to improve the accuracy of the rainfall estimation (Equation (20)).

$$R_{ENS} = C_1 R(Z_H) + C_2 R(K_{DP}) + C_3 R(Z_H, Z_{DR}) + C_4 R(Z_H, K_{DP}) + C_5 R(Z_{DR}, K_{DP}) + C_6 R(Z_H, Z_{DR}, K_{DP}) \qquad (20)$$

To calculate the coefficients of the relational expressions using the linear combination, coefficient optimization using HSA was performed, an error value (RMSE) compared the radar rainfall fields that the ensemble rainfall estimation relationship was applied to, and the ground AWS rain rate value was selected as a reference variable for optimization.

The raindrop size distribution characteristics affecting the parameters of the rainfall relationships change with the intensity of the rain rate [45–47]. Therefore, when optimizing the rainfall intensity using the HSA, the optimization was performed by dividing six rain rate categories (R1: $0.1 \leq R < 1.0$ mm h$^{-1}$; R2: $1.0 \leq R < 5.0$ mm h$^{-1}$; R3: $5.0 \leq R < 10.0$ mm h$^{-1}$; R4: $10.0 \leq R < 20.0$ mm h$^{-1}$; R5: $20.0 \leq R < 30.0$ mm h$^{-1}$; R6: $30.0 \leq R$). The HSA is a method that can be used to determine the optimal coefficient by repeating the process of applying an arbitrary value to the relational expression. The RMSE reduction results according to the number of rainfall estimation relationship repetitions determined using actual radar data are shown in Figure 4. When the number of repetitions was more than 250, the decrease in the RMSE decreased sharply, and when the number of repetitions was more than 600, the RMSE value converged to approximately 17 mm h$^{-1}$. As a result, the initial RMSE value of approximately 60 mm h$^{-1}$ decreased significantly to less than 20 mm h$^{-1}$.

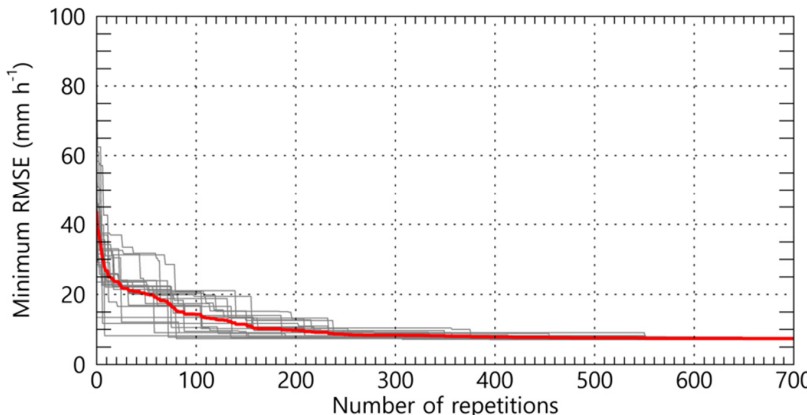

**Figure 4.** The convergence process of the RMSE with the number of repetitions using the harmony search algorithm. The gray and red lines indicate the HSA process results and the averaged RMSE results, respectively.

Figure 5 shows the rainfall estimation weight value for each rain rate intensity category using HSA. In the R1 and R2 intervals ($0.1 \leq R < 5$ mm h$^{-1}$), the weights of the R($Z_H$) and R($Z_H$, $Z_{DR}$), not considering $K_{DP}$, were significantly higher than the weights of the R($K_{DP}$) and R($Z_{DR}$, $K_{DP}$) relationships (Figure 5a,b). On the other hand, at rain rate intensities higher than 10 mm h$^{-1}$, the weighting factor values of R($K_{DP}$) and R($Z_{DR}$, $K_{DP}$) accounted for an increased maximum weight of 0.8 or more (Figure 5e,f). Therefore, the final rain rate fields were calculated by adjusting the weight values of the rainfall estimation relationships according to the rainfall intensity.

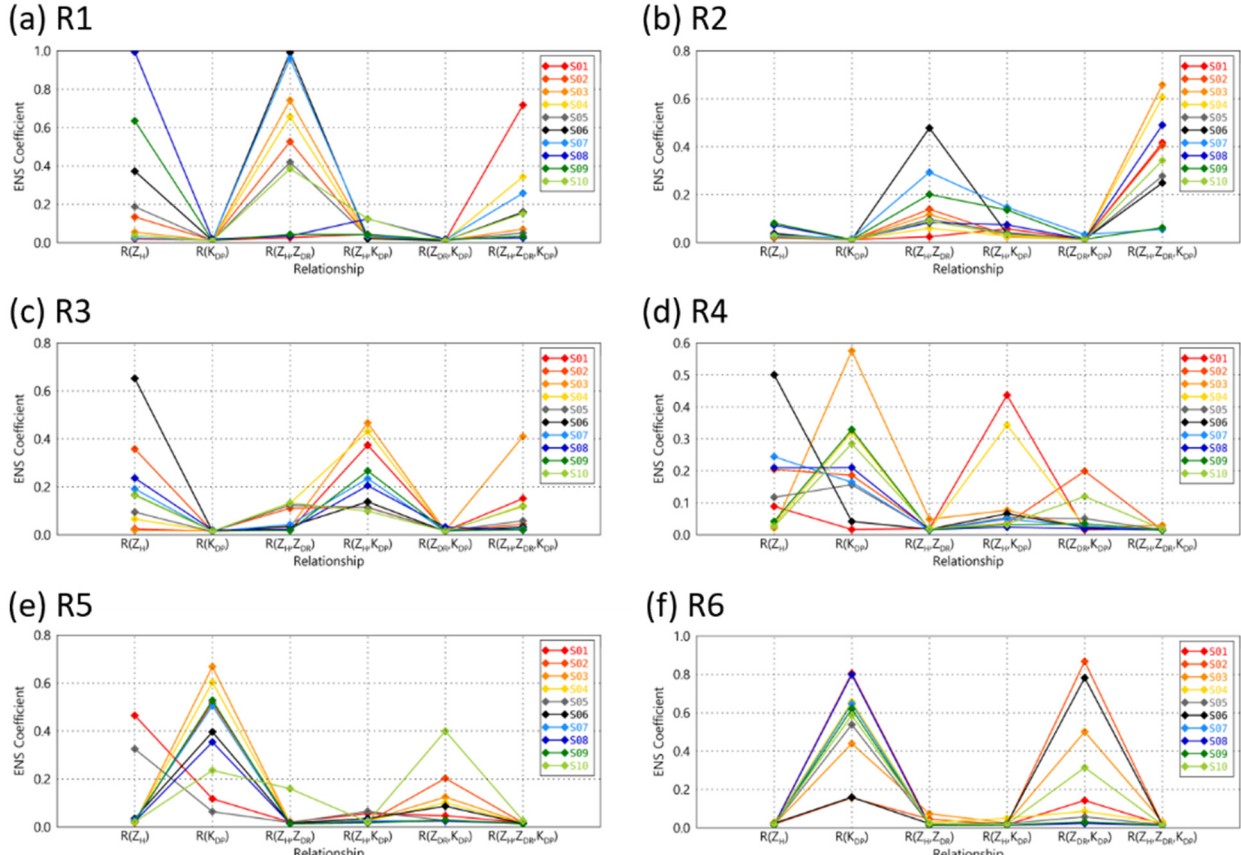

**Figure 5.** The ensemble coefficients of the QPE relationships for the rain rate categories at the 10 Parsivel sites.

The $R_{ENS}$ values obtained through the ensemble of rainfall estimation relationships and the RMSE, NE, and CORR results of each rainfall estimation relationship during the spring, summer, and fall season from 7 September 2018 to 30 November 2020 are shown in Figure 6. Overall, the error (i.e., RMSE and NE) values for the ensemble rainfall estimation relationships were lower than those of the independent rainfall estimation relationships, and the correlation with the ground observations was high. The relational expressions with high accuracy at all of the observation sites appeared in the following order: $R_{ENS}$, $R(Z_H,$ $K_{DP})$, $R(Z_H)$, $R(Z_H, Z_{DR})$, $R(Z_H, Z_{DR},$ and $K_{DP})$, $R(Z_{DR}, K_{DP})$, and $R(K_{DP})$. The correlation with the AWS observations appeared in the reverse order to the error accuracy.

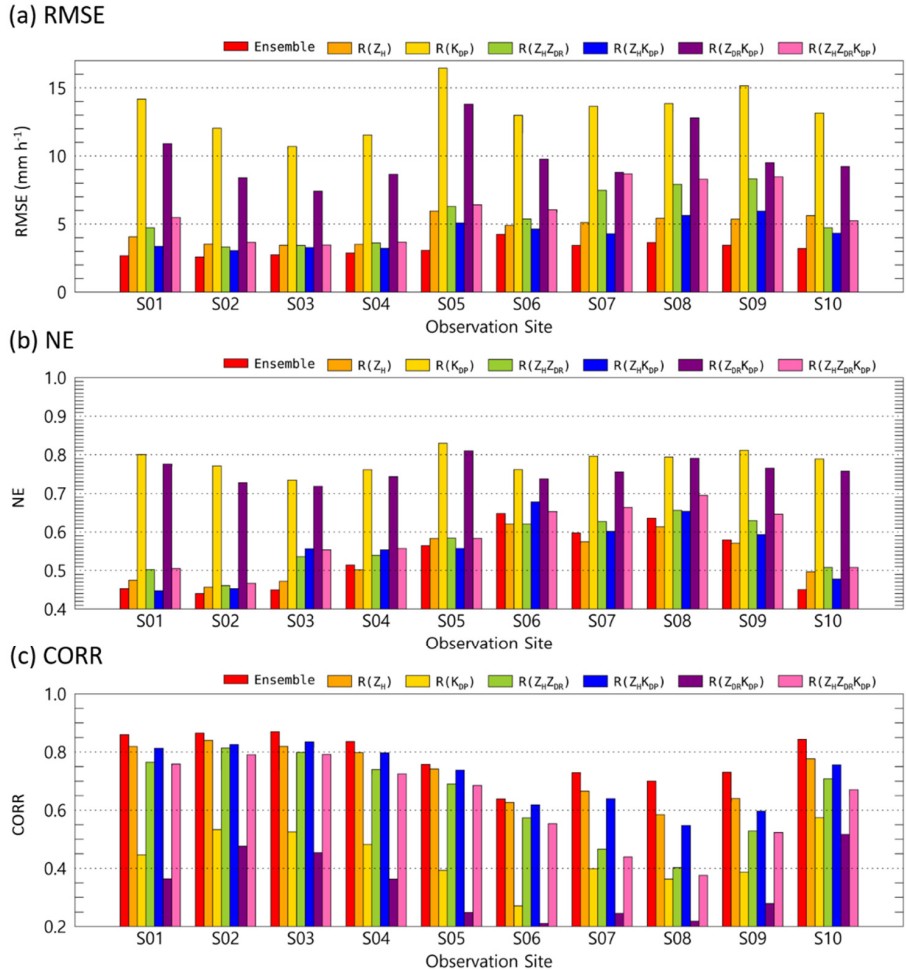

**Figure 6.** (**a**) RMSE, (**b**) NE, and (**c**) CORR of the ensemble results and QPE relationships at the 10 Parsivel sites.

To compare the rain rate intensity to the movement of the rainfall system, a time series analysis of the rain rate intensity was conducted on the precipitation cases on Jeju Island on 13 June 2020 (Figure 7). The selected precipitation case was a precipitation case in which the maximum daily cumulative rainfall in the Jeju area was 100 mm.

The rainfall estimation relationships, including the $K_{DP}$, showed an overall tendency for estimation, and the result was overestimated with a large width of up to 20 mm h$^{-1}$, even at weak rainfall intensities of less than 5 mm h$^{-1}$. For example, at 0640 LST, which was the time at which an extreme rain rate intensity was observed in a precipitation cell, the rainfall intensity that was estimated using the independent rainfall estimation relationship at S1 showed a maximum difference of approximately 80 mm h$^{-1}$ compared to the actual observed rain rate intensity (Figure 7a,c). However, as a result of the ensemble, the difference observed in the rain rate could be reduced to less than about 10 mm h$^{-1}$. In

particular, at S4–S10, the ensemble rain rate value recorded similar values when compared to the actual value, and the change in the increasing and decreasing rain rate intensity within a short period of time was estimated with a high level of accuracy (Figure 7d–j).

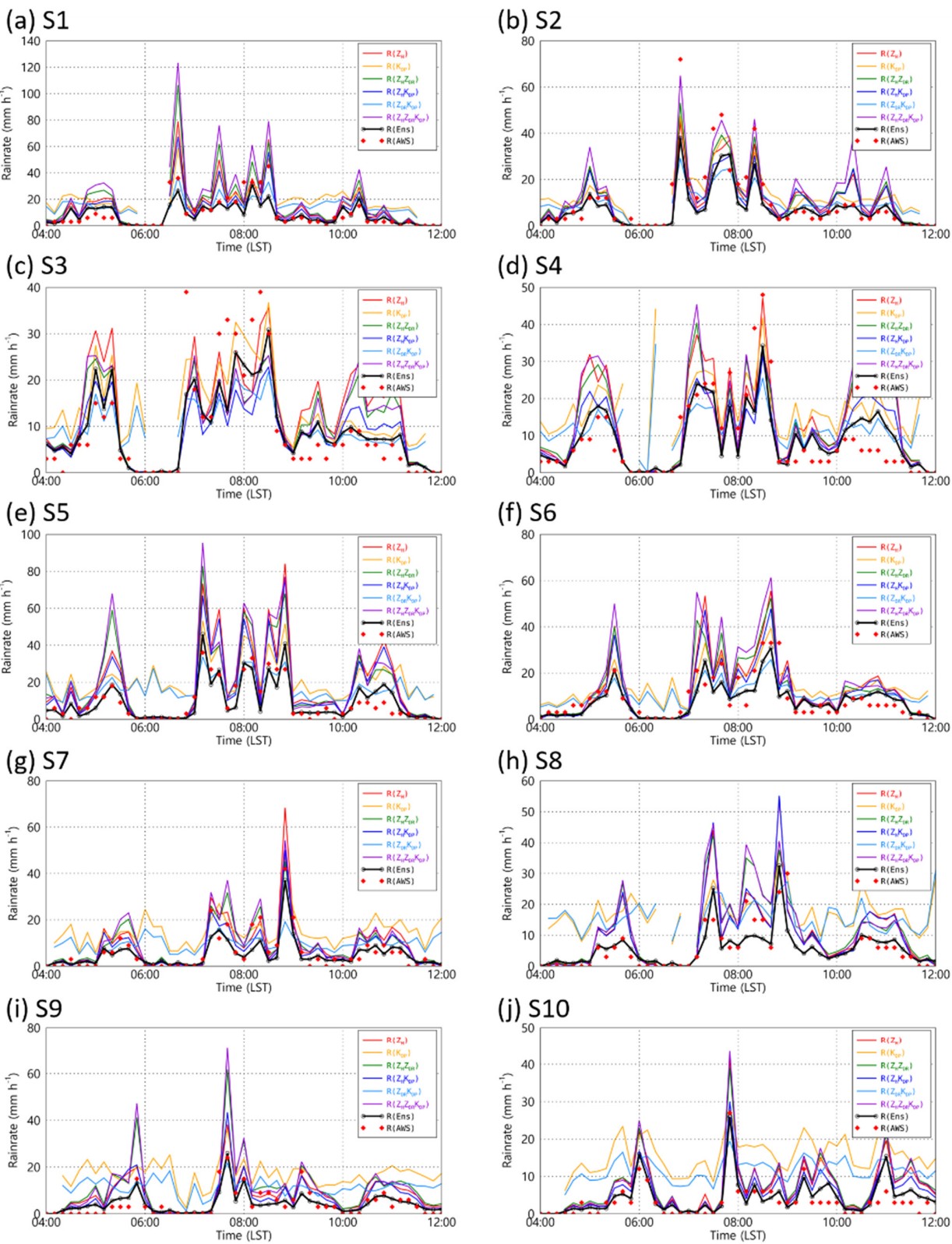

**Figure 7.** The time series of the rain rate calculated by the ensemble method (black, solid line), QPE relationships (colored, solid lines), and AWS (red diamond).

### 3.3. Spatial Correction of the Rainfall Fields

The left and right graphs in Figure 8 show the covariance matrix and decomposed triangular matrix for each AWS observation site for the error of the radar rain rate calculated using the rainfall estimation relationship obtained at S1 and the rain rate obtained from ground AWS observations. The larger the covariance, the higher the positive correlation between the two types of data compared, and the higher the error correlation with the observed values.

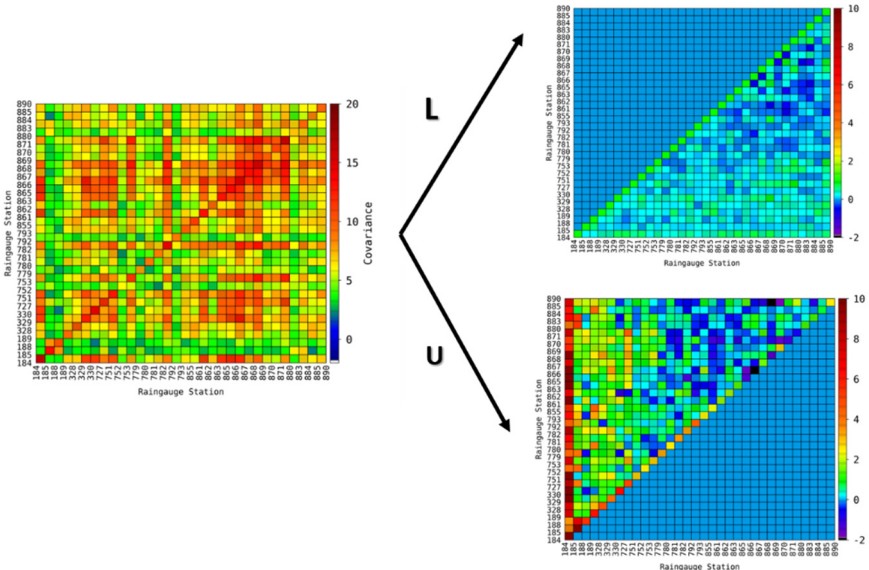

**Figure 8.** An example showing the decomposition of the lower and upper triangular matrix from the covariance matrix of the AWS rain rate.

The perturbation value of the rainfall fields to which the relationship of S6, which was the site with the highest altitude among the observation sites, was applied, recorded a low value of approximately 0.5 dBR in the mountainous peak areas but a negative perturbation value in the low-altitude areas close to the coastal line. Therefore, this result was underestimated compared to the actual observed value (Figure 9f). The spatial distribution characteristics for perturbation showed similar results according to the difference in the raindrop size distribution characteristics.

The perturbation distributions of S5 and S9, which were located on the leeward mountain slope, showed positive values at the mountainous peaks, and negative perturbation values were observed, resulting in underestimation at altitudes below 50 m ASL. The rainfall fields obtained at the sites located in the middle of the mountain slopes (i.e., S2, S4, S5, S7, S8, and S9), showed different perturbation values based on altitude rather than based on their relative position (windward/leeward) on the mountain.

Figure 9 shows the spatial perturbation distribution of the rainfall fields obtained through the ensemble of rainfall estimation relationships for each Parsivel observation site. The rainfall fields generated through the rainfall estimation relations obtained in coastal areas (i.e., S1 and S10) recorded high values of more than 2 dBR of perturbation in the mountainous regions (Figure 9a,j).

The perturbation distribution characteristics determined according to mountain altitude in Figure 9 are also shown in the comparison results of the rain rate values obtained from the rainfall fields using the ensemble rainfall estimation relationship and the ground AWS rain rate (Figure 10). The rainfall estimation relationships obtained in coastal areas had a very high error value (RMSE), higher than 7 mm h$^{-1}$ in mountainous regions, and had low error values of 3 mm h$^{-1}$ in coastal regions (Figure 10a,j).

The rainfall estimation relationship obtained at site S6 showed a relatively small value of approximately 7 mm h$^{-1}$ at a high altitude close to the mountain's peak. On the contrary,

the RMSE value in the coastal areas and on the areas of the mountain slope increased by approximately 3 mm h$^{-1}$ compared to the rainfall estimation relationship results that were obtained on the mountain slope and in the coastal areas. The final rainfall fields were obtained by averaging the spatially corrected rainfall fields at sites S1–S10 at the same grid point, and the comparison result with the observed rain rate on the ground is shown in Figure 11. By correcting the underestimated and overestimated errors according to the altitudes of the different areas on the mountain, the RMSE value for the area closest to the mountain peak decreased from 8 to approximately 6 mm h$^{-1}$, and the RMSE value in the coastal regions was kept as low as 3 mm h$^{-1}$.

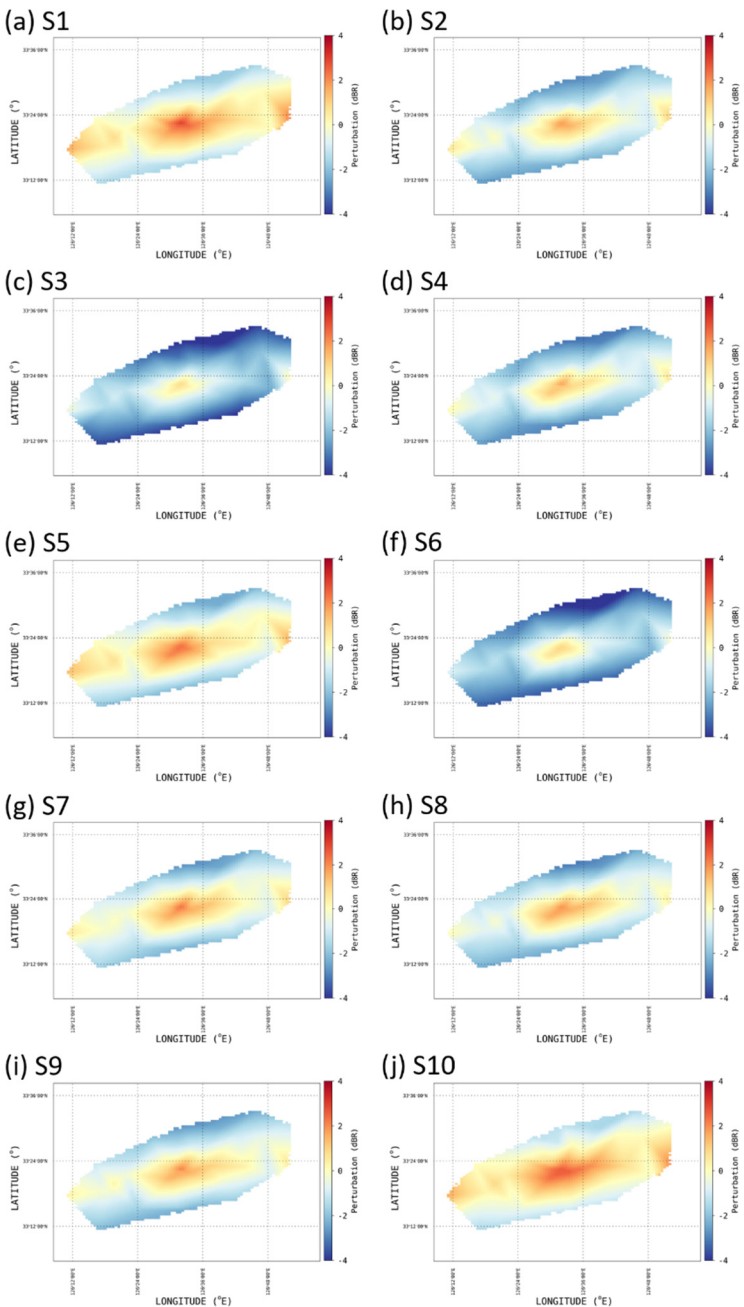

**Figure 9.** The perturbation fields at the 10 Parsivel sites.

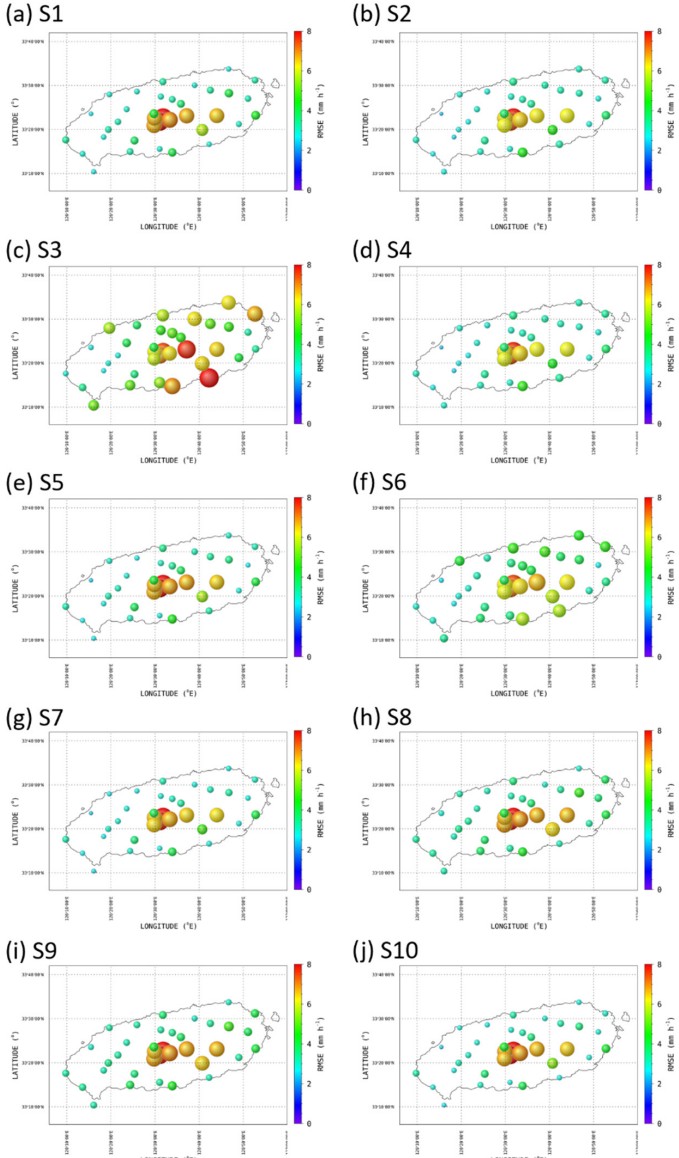

**Figure 10.** Bubble plots of the RMSEs of the ensemble QPE relationship calculated at the 10 Parsivel sites.

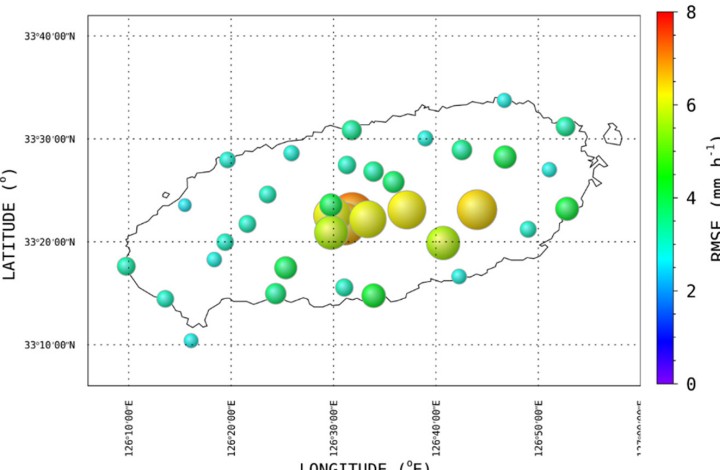

**Figure 11.** A bubble plot for the RMSEs of the corrected spatial error values for the rain rate fields.

## 4. Summary and Conclusions

The orographic rainfall was estimated using the DSD data obtained from in situ Parsivel observations. Polarization radar parameters, such as $Z_H$, $Z_{DR}$, and $K_{DP}$, were retrieved by DSD data obtained from the Parsivel observation data using T-matrix scattering simulation, and the rainfall estimation relationship was calculated. In addition, GSN and SSP dual-polarization radar data were used to verify the accuracy of the rainfall estimation relationships, and the lowest elevation angle observation data above the terrain elevation were used to minimize the beam blocking effect caused by the terrain.

The ensemble rainfall relationship was obtained via the linear combination of the rainfall relationships through the harmony search algorithm, and the error rate of the rainfall estimation relationship could be sufficiently reduced, even with 600 repetitions.

The rainfall relationship that included the $K_{DP}$ showed high accuracy for substantial rainfall events of over 20 mm h$^{-1}$. At weak rain rate intensities of less than 5 mm h$^{-1}$, the error was higher than that of the rainfall relationships that included the $Z_H$ and $Z_{DR}$ parameters. Therefore, the proportions of R($Z_H$) and R($Z_H$, $Z_{DR}$) were applied in rain rates lower than 5 mm h$^{-1}$, and the proportions of R($K_{DP}$) and R($Z_{DR}$, $K_{DP}$) were high at rain rates that were higher than 30 mm h$^{-1}$ when determining the ensemble rainfall relationship.

Spatial correction was applied to the rainfall fields to improve the accuracy of the rainfall fields over complex orographic rainfall. The spatial error characteristics with the altitude of the observation were applied to calculate the rainfall relationship. The rainfall generated through the relationship of obtained areas with a relatively high altitude had a relatively low error rate in sloped mountain slope areas, and it was underestimated in low-altitude areas and close to the sea. On the other hand, the rainfall generated from the coastal area showed overestimation in sloped mountain areas. In all of the coastal areas, the perturbation value was used as a correction factor for rainfall fields with a low recorded value that was between approximately −1 and 1 dBR. The spatial error difference had a more significant effect on the altitude than the windward/leeward side in the mountainous areas.

By optimizing rainfall estimation using HSA and the spatial rainfall error correction technique, it was possible to reduce the error rate according to the rainfall intensity of the rainfall relationship and the spatial errors caused by orographic development. Applying the rainfall field generation method proposed in this study to coastal and urban areas with complex topographical characteristics could help to improve the rainfall estimation accuracy for localized torrential rainfall.

**Author Contributions:** Conceptualization, C.-H.Y., H.-J.K. and S.-H.S.; investigation, S.-H.S. and W.J.; writing—original draft preparation, C.-H.Y. and H.-J.K.; writing—review and editing, C.-H.Y. and D.-I.L.; funding acquisition, C.-H.Y. All authors have read and agreed to the published version of the manuscript.

**Funding:** This research was supported by Basic Science Research Program through the National Research Foundation of Korea (NRF) funded by the Ministry of Education (NRF-2020R1I1A3066504).

**Institutional Review Board Statement:** Not applicable.

**Informed Consent Statement:** Not applicable.

**Data Availability Statement:** Not applicable.

**Conflicts of Interest:** The authors declare no conflict of interest.

## Appendix A

**Table A1.** The rainfall estimation relationships, RMSE, NE, and CORR considering the frequency of the GSN radar at each site.

| S1 | Equation | RMSE | NE | CORR |
|---|---|---|---|---|
| R($Z_H$) | R = 0.0389$Z_H^{0.65761}$ | 1.947 | 0.000041 | 0.945 |
| R($K_{DP}$) | R = 47.2049$K_{DP}^{0.72784}$ | 1.068 | 0.000031 | 0.981 |

**Table A1.** *Cont.*

| S1 | Equation | RMSE | NE | CORR |
|---|---|---|---|---|
| $R(Z_H, Z_{DR})$ | $R = 0.042Z_H{}^{0.7008}Z_{DR}{}^{-4.5108}$ | 1.893 | 0.000039 | 0.976 |
| $R(Z_H, K_{DP})$ | $R = 0.011Z_H{}^{0.7542}K_{DP}{}^{-0.1536}$ | 1.858 | 0.000042 | 0.938 |
| $R(Z_{DR}, K_{DP})$ | $R = 35.918Z_{DR}{}^{-1.0523}K_{DP}{}^{0.6414}$ | 2.035 | 0.000045 | 0.976 |
| $R(Z_H, Z_{DR}, K_{DP})$ | $R = 0.005Z_H{}^{0.9129}Z_{DR}{}^{-4.8768}K_{DP}{}^{-0.2149}$ | 1.836 | 0.000038 | 0.977 |
| **S2** | **Equation** | **RMSE** | **NE** | **CORR** |
| $R(Z_H)$ | $R = 0.0357Z_H{}^{0.67169}$ | 1.707 | 0.000046 | 0.967 |
| $R(K_{DP})$ | $R = 50.8550K_{DP}{}^{0.74610}$ | 1.065 | 0.000035 | 0.986 |
| $R(Z_H, Z_{DR})$ | $R = 0.040Z_H{}^{0.7015}Z_{DR}{}^{-3.8854}$ | 2.388 | 0.000048 | 0.977 |
| $R(Z_H, K_{DP})$ | $R = 0.008Z_H{}^{0.7879}K_{DP}{}^{-0.1773}$ | 1.859 | 0.000049 | 0.961 |
| $R(Z_{DR}, K_{DP})$ | $R = 31.234Z_{DR}{}^{-0.0326}K_{DP}{}^{0.6281}$ | 2.560 | 0.000056 | 0.974 |
| $R(Z_H, Z_{DR}, K_{DP})$ | $R = 0.004Z_H{}^{0.9269}Z_{DR}{}^{-4.2899}K_{DP}{}^{-0.2275}$ | 2.299 | 0.000046 | 0.979 |
| **S3** | **Equation** | **RMSE** | **NE** | **CORR** |
| $R(Z_H)$ | $R = 0.0291Z_H{}^{0.68783}$ | 1.560 | 0.000072 | 0.964 |
| $R(K_{DP})$ | $R = 49.9538K_{DP}{}^{0.75820}$ | 0.887 | 0.000053 | 0.985 |
| $R(Z_H, Z_{DR})$ | $R = 0.047Z_H{}^{0.6553}Z_{DR}{}^{-3.2141}$ | 2.442 | 0.000084 | 0.966 |
| $R(Z_H, K_{DP})$ | $R = 0.104Z_H{}^{0.5251}K_{DP}{}^{0.0651}$ | 1.880 | 0.000082 | 0.968 |
| $R(Z_{DR}, K_{DP})$ | $R = 31.641Z_{DR}{}^{-0.7763}K_{DP}{}^{0.6341}$ | 2.424 | 0.000086 | 0.968 |
| $R(Z_H, Z_{DR}, K_{DP})$ | $R = 0.023Z_H{}^{0.7233}Z_{DR}{}^{-3.3842}K_{DP}{}^{-0.0685}$ | 2.439 | 0.000084 | 0.966 |
| **S4** | **Equation** | **RMSE** | **NE** | **CORR** |
| $R(Z_H)$ | $R = 0.0369Z_H{}^{0.66180}$ | 2.721 | 0.000033 | 0.919 |
| $R(K_{DP})$ | $R = 48.5628K_{DP}{}^{0.73833}$ | 1.462 | 0.000025 | 0.970 |
| $R(Z_H, Z_{DR})$ | $R = 0.047Z_H{}^{0.6855}Z_{DR}{}^{-4.6863}$ | 2.916 | 0.000031 | 0.945 |
| $R(Z_H, K_{DP})$ | $R = 0.079Z_H{}^{0.5559}K_{DP}{}^{0.0289}$ | 2.294 | 0.000033 | 0.929 |
| $R(Z_{DR}, K_{DP})$ | $R = 37.293Z_{DR}{}^{-1.6821}K_{DP}{}^{0.6426}$ | 2.961 | 0.000033 | 0.952 |
| $R(Z_H, Z_{DR}, K_{DP})$ | $R = 0.011Z_H{}^{0.8256}Z_{DR}{}^{-5.0646}K_{DP}{}^{-0.1396}$ | 2.900 | 0.000031 | 0.945 |
| **S5** | **Equation** | **RMSE** | **NE** | **CORR** |
| $R(Z_H)$ | $R = 0.0575Z_H{}^{0.64395}$ | 2.471 | 0.000059 | 0.945 |
| $R(K_{DP})$ | $R = 56.5972K_{DP}{}^{0.71370}$ | 1.379 | 0.000044 | 0.979 |
| $R(Z_H, Z_{DR})$ | $R = 0.043Z_H{}^{0.7528}Z_{DR}{}^{-7.4426}$ | 2.649 | 0.000050 | 0.959 |
| $R(Z_H, K_{DP})$ | $R = 0.030Z_H{}^{0.6928}K_{DP}{}^{-0.0582}$ | 2.165 | 0.000055 | 0.941 |
| $R(Z_{DR}, K_{DP})$ | $R = 58.388Z_{DR}{}^{-3.7367}K_{DP}{}^{0.6854}$ | 2.995 | 0.000060 | 0.968 |
| $R(Z_H, Z_{DR}, K_{DP})$ | $R = 0.008Z_H{}^{0.9155}Z_{DR}{}^{-7.8030}K_{DP}{}^{-0.1609}$ | 2.580 | 0.000049 | 0.961 |
| **S6** | **Equation** | **RMSE** | **NE** | **CORR** |
| $R(Z_H)$ | $R = 0.0404Z_H{}^{0.66853}$ | 1.918 | 0.000026 | 0.960 |
| $R(K_{DP})$ | $R = 52.4869K_{DP}{}^{0.72923}$ | 1.148 | 0.000019 | 0.984 |
| $R(Z_H, Z_{DR})$ | $R = 0.043Z_H{}^{0.7192}Z_{DR}{}^{-5.1591}$ | 2.203 | 0.000025 | 0.981 |
| $R(Z_H, K_{DP})$ | $R = 0.017Z_H{}^{0.7312}K_{DP}{}^{-0.1140}$ | 1.891 | 0.000026 | 0.957 |
| $R(Z_{DR}, K_{DP})$ | $R = 36.494Z_{DR}{}^{-0.7470}K_{DP}{}^{0.6334}$ | 2.507 | 0.000031 | 0.979 |
| $R(Z_H, Z_{DR}, K_{DP})$ | $R = 0.008Z_H{}^{0.8891}Z_{DR}{}^{-5.5056}K_{DP}{}^{-0.1682}$ | 2.122 | 0.000024 | 0.982 |
| **S7** | **Equation** | **RMSE** | **NE** | **CORR** |
| $R(Z_H)$ | $R = 0.0371Z_H{}^{0.67122}$ | 1.077 | 0.000064 | 0.956 |
| $R(K_{DP})$ | $R = 50.4189K_{DP}{}^{0.73234}$ | 0.872 | 0.000055 | 0.969 |
| $R(Z_H, Z_{DR})$ | $R = 0.042Z_H{}^{0.7460}Z_{DR}{}^{-8.1627}$ | 1.996 | 0.000060 | 0.894 |
| $R(Z_H, K_{DP})$ | $R = 0.015Z_H{}^{0.7230}K_{DP}{}^{-0.1251}$ | 1.206 | 0.000065 | 0.952 |
| $R(Z_{DR}, K_{DP})$ | $R = 32.844Z_{DR}{}^{-2.8589}K_{DP}{}^{0.5979}$ | 2.221 | 0.000077 | 0.904 |
| $R(Z_H, Z_{DR}, K_{DP})$ | $R = 0.007Z_H{}^{0.9252}Z_{DR}{}^{-8.4534}K_{DP}{}^{-0.1701}$ | 1.919 | 0.000057 | 0.903 |
| **S8** | **Equation** | **RMSE** | **NE** | **CORR** |
| $R(Z_H)$ | $R = 0.0504Z_H{}^{0.64806}$ | 2.485 | 0.000036 | 0.926 |
| $R(K_{DP})$ | $R = 52.0774K_{DP}{}^{0.70816}$ | 1.332 | 0.000029 | 0.972 |
| $R(Z_H, Z_{DR})$ | $R = 0.042Z_H{}^{0.7653}Z_{DR}{}^{-7.9757}$ | 2.200 | 0.000027 | 0.944 |

**Table A1.** *Cont.*

| S8 | Equation | RMSE | NE | CORR |
|---|---|---|---|---|
| $R(Z_H, K_{DP})$ | $R = 0.014Z_H^{0.7618}K_{DP}^{-0.1394}$ | 2.452 | 0.000035 | 0.915 |
| $R(Z_{DR}, K_{DP})$ | $R = 58.268Z_{DR}^{-4.0993}K_{DP}^{0.6685}$ | 2.596 | 0.000036 | 0.946 |
| $R(Z_H, Z_{DR}, K_{DP})$ | $R = 0.007Z_H^{0.9357}Z_{DR}^{-8.1194}K_{DP}^{-0.1700}$ | 2.109 | 0.000026 | 0.949 |
| **S9** | **Equation** | **RMSE** | **NE** | **CORR** |
| $R(Z_H)$ | $R = 0.1181Z_H^{0.55311}$ | 2.539 | 0.000115 | 0.945 |
| $R(K_{DP})$ | $R = 50.3939K_{DP}^{0.64553}$ | 2.031 | 0.000111 | 0.968 |
| $R(Z_H, Z_{DR})$ | $R = 0.052Z_H^{0.7473}Z_{DR}^{-7.9144}$ | 5.608 | 0.000113 | 0.813 |
| $R(Z_H, K_{DP})$ | $R = 0.009Z_H^{0.7997}K_{DP}^{-0.2066}$ | 3.240 | 0.000110 | 0.933 |
| $R(Z_{DR}, K_{DP})$ | $R = 34.728Z_{DR}^{-2.9961}K_{DP}^{0.5528}$ | 6.163 | 0.000142 | 0.809 |
| $R(Z_H, Z_{DR}, K_{DP})$ | $R = 0.005Z_H^{0.9754}Z_{DR}^{-8.0061}K_{DP}^{-0.2175}$ | 5.269 | 0.000105 | 0.851 |
| **S10** | **Equation** | **RMSE** | **NE** | **CORR** |
| $R(Z_H)$ | $R = 0.0430Z_H^{0.63962}$ | 1.469 | 0.000047 | 0.955 |
| $R(K_{DP})$ | $R = 43.3704K_{DP}^{0.70726}$ | 0.876 | 0.000036 | 0.979 |
| $R(Z_H, Z_{DR})$ | $R = 0.055Z_H^{0.6415}Z_{DR}^{-3.5784}$ | 1.993 | 0.000043 | 0.953 |
| $R(Z_H, K_{DP})$ | $R = 0.047Z_H^{0.5990}K_{DP}^{-0.0292}$ | 1.408 | 0.000046 | 0.957 |
| $R(Z_{DR}, K_{DP})$ | $R = 29.611Z_{DR}^{-0.7941}K_{DP}^{0.6104}$ | 1.967 | 0.000047 | 0.961 |
| $R(Z_H, Z_{DR}, K_{DP})$ | $R = 0.017Z_H^{0.7560}Z_{DR}^{-3.8443}K_{DP}^{-0.1160}$ | 1.980 | 0.000043 | 0.953 |

**Table A2.** The rainfall estimation relationships considering the frequency of the SSP radar at each site.

| S1 | Equation | RMSE | NE | CORR |
|---|---|---|---|---|
| $R(Z_H)$ | $R = 0.0389Z_H^{0.65739}$ | 1.957 | 0.000041 | 0.944 |
| $R(K_{DP})$ | $R = 46.3053K_{DP}^{0.72766}$ | 1.070 | 0.000031 | 0.981 |
| $R(Z_H, Z_{DR})$ | $R = 0.042Z_H^{0.7008}Z_{DR}^{-4.5119}$ | 1.893 | 0.000039 | 0.976 |
| $R(Z_H, K_{DP})$ | $R = 0.011Z_H^{0.7526}K_{DP}^{-0.1522}$ | 1.863 | 0.000042 | 0.938 |
| $R(Z_{DR}, K_{DP})$ | $R = 35.344Z_{DR}^{-1.0554}K_{DP}^{0.6414}$ | 2.035 | 0.000045 | 0.976 |
| $R(Z_H, Z_{DR}, K_{DP})$ | $R = 0.005Z_H^{0.9115}Z_{DR}^{-4.8764}K_{DP}^{-0.2136}$ | 1.836 | 0.000038 | 0.977 |
| **S2** | **Equation** | **RMSE** | **NE** | **CORR** |
| $R(Z_H)$ | $R = 0.0357Z_H^{0.67142}$ | 1.711 | 0.000047 | 0.967 |
| $R(K_{DP})$ | $R = 49.8511K_{DP}^{0.74583}$ | 1.066 | 0.000035 | 0.986 |
| $R(Z_H, Z_{DR})$ | $R = 0.040Z_H^{0.7014}Z_{DR}^{-3.8879}$ | 2.388 | 0.000048 | 0.977 |
| $R(Z_H, K_{DP})$ | $R = 0.008Z_H^{0.7886}K_{DP}^{-0.1783}$ | 1.861 | 0.000049 | 0.961 |
| $R(Z_{DR}, K_{DP})$ | $R = 30.906Z_{DR}^{-0.0570}K_{DP}^{0.6289}$ | 2.559 | 0.000055 | 0.974 |
| $R(Z_H, Z_{DR}, K_{DP})$ | $R = 0.004Z_H^{0.9283}Z_{DR}^{-4.2948}K_{DP}^{-0.2290}$ | 2.299 | 0.000046 | 0.979 |
| **S3** | **Equation** | **RMSE** | **NE** | **CORR** |
| $R(Z_H)$ | $R = 0.0291Z_H^{0.68757}$ | 1.566 | 0.000072 | 0.964 |
| $R(K_{DP})$ | $R = 48.9830K_{DP}^{0.75822}$ | 0.888 | 0.000053 | 0.985 |
| $R(Z_H, Z_{DR})$ | $R = 0.047Z_H^{0.6552}Z_{DR}^{-3.2163}$ | 2.442 | 0.000084 | 0.966 |
| $R(Z_H, K_{DP})$ | $R = 0.105Z_H^{0.5238}K_{DP}^{0.0664}$ | 1.880 | 0.000082 | 0.968 |
| $R(Z_{DR}, K_{DP})$ | $R = 31.152Z_{DR}^{-0.7808}K_{DP}^{0.6341}$ | 2.424 | 0.000086 | 0.968 |
| $R(Z_H, Z_{DR}, K_{DP})$ | $R = 0.023Z_H^{0.7239}Z_{DR}^{-3.3888}K_{DP}^{-0.0692}$ | 2.438 | 0.000084 | 0.966 |
| **S4** | **Equation** | **RMSE** | **NE** | **CORR** |
| $R(Z_H)$ | $R = 0.0369Z_H^{0.66152}$ | 2.736 | 0.000033 | 0.918 |
| $R(K_{DP})$ | $R = 47.6293K_{DP}^{0.73827}$ | 1.465 | 0.000025 | 0.970 |
| $R(Z_H, Z_{DR})$ | $R = 0.047Z_H^{0.6854}Z_{DR}^{-4.6859}$ | 2.916 | 0.000031 | 0.945 |
| $R(Z_H, K_{DP})$ | $R = 0.080Z_H^{0.5545}K_{DP}^{0.0301}$ | 2.296 | 0.000033 | 0.929 |
| $R(Z_{DR}, K_{DP})$ | $R = 36.783Z_{DR}^{-1.6926}K_{DP}^{0.6429}$ | 2.960 | 0.000033 | 0.952 |
| $R(Z_H, Z_{DR}, K_{DP})$ | $R = 0.011Z_H^{0.8249}Z_{DR}^{-5.0622}K_{DP}^{-0.1391}$ | 2.901 | 0.000031 | 0.945 |
| **S5** | **Equation** | **RMSE** | **NE** | **CORR** |
| $R(Z_H)$ | $R = 0.0575Z_H^{0.64375}$ | 2.479 | 0.000059 | 0.945 |

**Table A2.** *Cont.*

| S5 | Equation | RMSE | NE | CORR |
|---|---|---|---|---|
| $R(K_{DP})$ | $R = 55.5569K_{DP}^{0.71362}$ | 1.381 | 0.000044 | 0.979 |
| $R(Z_H, Z_{DR})$ | $R = 0.043Z_H^{0.7527}Z_{DR}^{-7.4415}$ | 2.649 | 0.000050 | 0.959 |
| $R(Z_H, K_{DP})$ | $R = 0.030Z_H^{0.6923}K_{DP}^{-0.0579}$ | 2.171 | 0.000055 | 0.941 |
| $R(Z_{DR}, K_{DP})$ | $R = 57.343Z_{DR}^{-3.7328}K_{DP}^{0.6854}$ | 2.995 | 0.000060 | 0.968 |
| $R(Z_H, Z_{DR}, K_{DP})$ | $R = 0.008Z_H^{0.9154}Z_{DR}^{-7.8027}K_{DP}^{-0.1609}$ | 2.580 | 0.000049 | 0.961 |
| **S6** | **Equation** | **RMSE** | **NE** | **CORR** |
| $R(Z_H)$ | $R = 0.0404Z_H^{0.66833}$ | 1.924 | 0.000026 | 0.960 |
| $R(K_{DP})$ | $R = 51.4925K_{DP}^{0.72911}$ | 1.150 | 0.000019 | 0.983 |
| $R(Z_H, Z_{DR})$ | $R = 0.043Z_H^{0.7191}Z_{DR}^{-5.1602}$ | 2.203 | 0.000025 | 0.981 |
| $R(Z_H, K_{DP})$ | $R = 0.017Z_H^{0.7304}K_{DP}^{-0.1133}$ | 1.894 | 0.000026 | 0.956 |
| $R(Z_{DR}, K_{DP})$ | $R = 35.917Z_{DR}^{-0.7495}K_{DP}^{0.6334}$ | 2.506 | 0.000031 | 0.979 |
| $R(Z_H, Z_{DR}, K_{DP})$ | $R = 0.008Z_H^{0.8885}Z_{DR}^{-5.5065}K_{DP}^{-0.1678}$ | 2.122 | 0.000024 | 0.982 |
| **S7** | **Equation** | **RMSE** | **NE** | **CORR** |
| $R(Z_H)$ | $R = 0.0371Z_H^{0.67104}$ | 1.078 | 0.000064 | 0.956 |
| $R(K_{DP})$ | $R = 49.4729K_{DP}^{0.73230}$ | 0.872 | 0.000055 | 0.969 |
| $R(Z_H, Z_{DR})$ | $R = 0.042Z_H^{0.7459}Z_{DR}^{-8.1628}$ | 1.997 | 0.000060 | 0.894 |
| $R(Z_H, K_{DP})$ | $R = 0.015Z_H^{0.7234}K_{DP}^{-0.1256}$ | 1.206 | 0.000065 | 0.952 |
| $R(Z_{DR}, K_{DP})$ | $R = 32.427Z_{DR}^{-2.8688}K_{DP}^{0.5982}$ | 2.221 | 0.000077 | 0.904 |
| $R(Z_H, Z_{DR}, K_{DP})$ | $R = 0.007Z_H^{0.9257}Z_{DR}^{-8.4539}K_{DP}^{-0.1707}$ | 1.920 | 0.000057 | 0.903 |
| **S8** | **Equation** | **RMSE** | **NE** | **CORR** |
| $R(Z_H)$ | $R = 0.0504Z_H^{0.64787}$ | 2.499 | 0.000036 | 0.925 |
| $R(K_{DP})$ | $R = 51.1323K_{DP}^{0.70813}$ | 1.335 | 0.000029 | 0.972 |
| $R(Z_H, Z_{DR})$ | $R = 0.042Z_H^{0.7652}Z_{DR}^{-7.9749}$ | 2.201 | 0.000027 | 0.944 |
| $R(Z_H, K_{DP})$ | $R = 0.014Z_H^{0.7610}K_{DP}^{-0.1388}$ | 2.466 | 0.000035 | 0.915 |
| $R(Z_{DR}, K_{DP})$ | $R = 57.234Z_{DR}^{-4.0948}K_{DP}^{0.6684}$ | 2.596 | 0.000036 | 0.946 |
| $R(Z_H, Z_{DR}, K_{DP})$ | $R = 0.007Z_H^{0.9353}Z_{DR}^{-8.1196}K_{DP}^{-0.1696}$ | 2.110 | 0.000026 | 0.949 |
| **S9** | **Equation** | **RMSE** | **NE** | **CORR** |
| $R(Z_H)$ | $R = 0.1183Z_H^{0.55280}$ | 2.545 | 0.000116 | 0.945 |
| $R(K_{DP})$ | $R = 49.5342K_{DP}^{0.64540}$ | 2.031 | 0.000111 | 0.968 |
| $R(Z_H, Z_{DR})$ | $R = 0.052Z_H^{0.7471}Z_{DR}^{-7.9075}$ | 5.609 | 0.000113 | 0.813 |
| $R(Z_H, K_{DP})$ | $R = 0.009Z_H^{0.7990}K_{DP}^{-0.2062}$ | 3.262 | 0.000111 | 0.932 |
| $R(Z_{DR}, K_{DP})$ | $R = 34.201Z_{DR}^{-2.9886}K_{DP}^{0.5526}$ | 6.163 | 0.000142 | 0.809 |
| $R(Z_H, Z_{DR}, K_{DP})$ | $R = 0.005Z_H^{0.9751}Z_{DR}^{-8.0012}K_{DP}^{-0.2173}$ | 5.271 | 0.000105 | 0.851 |
| **S10** | **Equation** | **RMSE** | **NE** | **CORR** |
| $R(Z_H)$ | $R = 0.0430Z_H^{0.63940}$ | 1.475 | 0.000047 | 0.955 |
| $R(K_{DP})$ | $R = 42.5642K_{DP}^{0.70709}$ | 0.877 | 0.000036 | 0.979 |
| $R(Z_H, Z_{DR})$ | $R = 0.055Z_H^{0.6415}Z_{DR}^{-3.5808}$ | 1.993 | 0.000043 | 0.953 |
| $R(Z_H, K_{DP})$ | $R = 0.047Z_H^{0.5995}K_{DP}^{-0.0298}$ | 1.408 | 0.000046 | 0.957 |
| $R(Z_{DR}, K_{DP})$ | $R = 29.116Z_{DR}^{-0.7905}K_{DP}^{0.6102}$ | 1.966 | 0.000047 | 0.961 |
| $R(Z_H, Z_{DR}, K_{DP})$ | $R = 0.017Z_H^{0.7572}Z_{DR}^{-3.8506}K_{DP}^{-0.1172}$ | 1.980 | 0.000043 | 0.953 |

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
