# Peer review of "Dual-Polarization Radar-Based Quantitative Precipitation Estimation of Mountain Terrain Using Multi-Disdrometer Data"

_remotesensing, doi:10.3390/rs14102290_

Round 1

Author Response

In order to satisfy all reviewers’ comments, the authors tried to correct the manuscript according to the comments. Please refer to the answers in response file for comments
In the following, the comments made by the referees appear in black, while our replies are in red, and the proposed modified text in the manuscript is in blue.
We believe our results and discussions should be of interest for readers. We hope your favorable considerations for publication of this paper to Remote Sensing.

Reviewer 2 Report

This is a very interesting work that it deserves for publications. However, I would like to make a few more moderate comments

First of all, I would like to mention that you have not include any vital information of the type of the radar, the type of the scans that it performs (i.e. PPI or volume scans or if it makes any RHI, etc.) how long it takes one full volume scan, what is the azimuth and range resolution of the radar, please give a full clear paragraph dedicated to the radar only. If you include a table with some technical information is also important.

Secondly, for the results, I feel like it is important to include a table with your statistical metrics that you present in Figure 3. I would prefere to see Tables 4 and 5 in an appendix senction but this is up to authors.

Now as far Figure 5 I am not sure I understood the gradidute of the weighting effect. Do you expect something better or worst with closer to 1 or 0 and why. Please elaborate a little bit more on your results.

I am not sure I have understand the subsection 3.3 and I am not sure I see radar rainfall estimation results illustrated on a Figure.

Please check also your English.

On top of that I would like to point out to the authors valuable reference that I feel it fits in this work:

Derin et al. 2018: 10.1109/TGRS.2017.2763622

Author Response

(The authors gave the same response as above.)

Reviewer 3 Report

This manuscript presents dual-polarization radar-based quantitative precipitation estimation of mountain terrain using multi-disdrometer data. It is a topic of interest to the researchers in the related areas. However, I personally think that the manuscript needs some minor improvement before acceptance for publication. My detailed comments are as follows:

(1) In Section 2.1.2, the constant in Equation 2 was set as 6, as suggested by Freidrich et al. But have the authors considered the applicability of this given constant? Also, the meaning of this constant is unclear.

(2) In Section 2.1.3, why was the set temperature variable assumed to be 20°C? Please give a brief basis for your assumption.

(3) In Figure 3, the authors did not explain the meaning of NE. Since it is one of the figure legends, NE should be given the necessary description.

(4) There are errors or omissions in individual references, e.g. in reference 2, the title and author information is wrong and the volume number is missing.

Author Response

(The authors gave the same response as above.)

Round 2

Reviewer 1 Report

Thanks to the authors for their efforts in improving the manuscript. I believe it deserves to be published in its present form.